# Autophagy sustains glutamate and aspartate synthesis in *Saccharomyces cerevisiae* during nitrogen starvation

Kuanqing Liu[1], Benjamin M. Sutter[1] & Benjamin P. Tu [1]✉

Autophagy catabolizes cellular constituents to promote survival during nutrient deprivation. Yet, a metabolic comprehension of this recycling operation, despite its crucial importance, remains incomplete. Here, we uncover a specific metabolic function of autophagy that exquisitely adjusts cellular metabolism according to nitrogen availability in the budding yeast *Saccharomyces cerevisiae*. Autophagy enables metabolic plasticity to promote glutamate and aspartate synthesis, which empowers nitrogen-starved cells to replenish their nitrogen currency and sustain macromolecule synthesis. Our findings provide critical insights into the metabolic basis by which autophagy recycles cellular components and may also have important implications in understanding the role of autophagy in diseases such as cancer.

---

[1] Department of Biochemistry, University of Texas Southwestern Medical Center, Dallas, TX, USA. ✉email: benjamin.tu@utsouthwestern.edu

Living organisms depend on a steady supply of nutrients to grow and proliferate. In their natural habitats, however, nutrient availability often fluctuates and occasionally can become limited or even absent. To cope with nutrient shortage, both prokaryotes and eukaryotes have evolved to harness multiple elaborate, albeit distinct mechanisms. One of them, operating pervasively in eukaryotes is autophagy, a "self-eating" process that delivers intracellular materials to the lysosome/vacuole for degradation[1]. Autophagy can be classified into three major types: macroautophagy, microautophagy, and chaperone-mediated autophagy (CMA)[1–3]. Macroautophagy (hereafter referred to as autophagy) is the most relevant type under nutrient starvation[4] and it relies on double-membrane vesicles termed autophagosomes to deliver intracellular cargos to the lysosome/vacuole for degradation. By contrast, microautophagy transports cellular contents via direct lysosomal/vacuolar invagination. CMA, on the other hand, facilitates the direct translocation of proteins across the lysosomal membrane in mammalian cells[5].

Autophagy is crucial for surviving nutrient starvation in various model organisms[6–15]. Although this pro-survival function is often attributed to its cannibalistic actions that presumably provide cells with the necessary nutrients to survive starvation[4,16], it remains incompletely understood how autophagy supports cellular metabolism to enhance survival under nutrient deprivation. Of the many cargos targeted by autophagy during starvation, proteins are pivotal; they are the most abundant intracellular macromolecules[17] and the derived amino acids can fulfill various important cellular functions[4,18]. Protein degradation by autophagy was initially observed in animal cells[19–21], and it was later confirmed in yeast cells following the discovery of the ATG genes[14]. Consistently, autophagy-deficient cells fail to maintain their amino acid pools during starvation and eventually succumb to death[6,9,11,13,22,23]. These studies, together with the identification of Atg22p as a vacuolar effluxer for amino acids[24], support a model where autophagy degrades proteins inside the lysosome/vacuole and the derived amino acids are exported to cope with starvation[16,18,22,24]. But how cells might reutilize the exported amino acids remains incompletely understood.

Here, we demonstrate that nitrogen-starved yeast cells do not simply reuse these amino acids as they are, but instead employ a plethora of transamination and deamination reactions to scavenge nitrogen to sustain glutamate synthesis. Amassing glutamate, the major cellular nitrogen donor[25], effectively replenishes cellular nitrogen currency and provides cells with the resources to reinvest in nitrogen anabolism, such as aspartate synthesis, to support macromolecule synthesis. Given that certain cancers can exploit autophagy as a crucial survival mechanism[26–28], our work may have important implications in understanding the molecular basis by which autophagy protects cancer cells from stress.

## Results
### Autophagy is critical for maintaining amino acid pools under nitrogen starvation.
To understand the metabolic functions of autophagy, we adopted a metabolomics-based approach and systematically investigated how nitrogen-starved yeast cells harness autophagy to degrade proteins and salvage amino acids. Prototrophic yeast strains were grown in synthetic defined (SD) medium and shifted to synthetic defined medium without nitrogen (SD-N) (Fig. 1a). As anticipated, this popular medium switch robustly induced autophagy in wild type (WT) cells, but not in autophagy-deficient atg1Δ cells (Supplementary Fig. 1a). By contrast, starvation for glucose, or glucose and nitrogen, failed to significantly stimulate autophagy (Supplementary Fig. 1b), as previously reported[29]. Phenotypically, WT and atg1Δ cells exhibited virtually indistinguishable growth in SD, but the former

became clearly more proficient at adapting to nitrogen starvation (Supplementary Figs. 1c and 1d), consistent with previous findings[14]. Taken together, these observations reaffirm the notion that autophagy functions as a critical stress response to cope with nutrient starvation.

We next measured changes of amino acids in yeast cells using liquid chromatography-tandem mass spectrometry. With nitrogen, WT and atg1Δ cells exhibited comparable profiles of amino acids, and intermediates of glycolysis and the tricarboxylic acid (TCA) cycle (Fig. 1b, and Supplementary Figs. 2a and 2b). Nitrogen starvation caused an abrupt surge of α-ketoglutarate and a precipitous decline of amino acids in both strains (Fig. 1b, and Supplementary Figs. 2a and 2b). However, WT cells were apparently more competent at preserving their amino acid levels than atg1Δ cells (Fig. 1b and Supplementary Fig. 3a). These observations reinforce the importance of autophagy in maintaining amino acid pools during nitrogen starvation, as has been reported previously[22,23].

Intriguingly, certain amino acids benefited from autophagy only marginally (e.g., methionine and serine), some modestly (e.g., phenylalanine, tyrosine, and isoleucine), whereas others substantially (e.g., aspartate, glutamate, valine, and glutamine) (Fig. 1b and Supplementary Fig. 3a). Since production and consumption ultimately determine the abundance of a metabolite, we reasoned that cells would have to modulate these two opposing processes to account for this peculiar disparity. For aspartate and glutamate, preferentially degrading acidic proteins via autophagy could explain their biased accumulation. However, this would not be compatible with the generally nonselective nature of autophagy[4]. Alternatively, reduced consumption could cause an accumulation of aspartate and glutamate. However, cells still require amino acids to support translation during nitrogen starvation[22]. These conundrums prompted us to consider the possibility that autophagy might promote the synthesis of specific amino acids under nitrogen starvation.

### Autophagy sustains glutamate and aspartate synthesis during nitrogen starvation.
To test this hypothesis, we chose to focus primarily on aspartate and glutamate, because of their striking dependence on autophagy and their abundance compared to other amino acids (Fig. 1b, and Supplementary Figs. 3a and 3b). We also verified that amassing aspartate and glutamate under nitrogen starvation requires an intact autophagy machinery, but not mitophagy, the cytoplasm to vacuole targeting pathway, or piecemeal microautophagy (Supplementary Fig. 3c). In addition, the differences in aspartate and glutamate levels between WT and atg1Δ cells are not sensitive to normalization methods: optical density at 600 nm (OD$_{600}$), which accounts for cell size change, and total protein content gave similar results (Supplementary Figs. 4a and 4b).

Next, we performed an isotope-tracing experiment where cells were grown in SD and switched to SD-N with uniformly [13]C labeled ([U-13]C) glucose (Fig. 2a). In prototrophic yeast, the 20 proteogenic amino acids derive their carbon skeletons, directly or indirectly, from glycolytic and TCA intermediates. Therefore, if there is de novo amino acid synthesis, with efficient [U-13]C-carbon assimilation (Supplementary Fig. 5a), the [U-13]C-species would become a dominant form for many newly synthesized amino acids. Accordingly, measuring the [U-12]C- and [U-13]C-species would be sufficient to capture the major changes of the amino acid pools. This assumption was indeed validated by the comparable sensitivities between [U-12]C- and [U-13]C-amino acids (Supplementary Fig. 5b) and the similar amino acid profiles between the standard and 13C-tracing experiments (Figs. 1b and 2b, and Supplementary Figs. 3a and 5c). Furthermore, we confirmed for

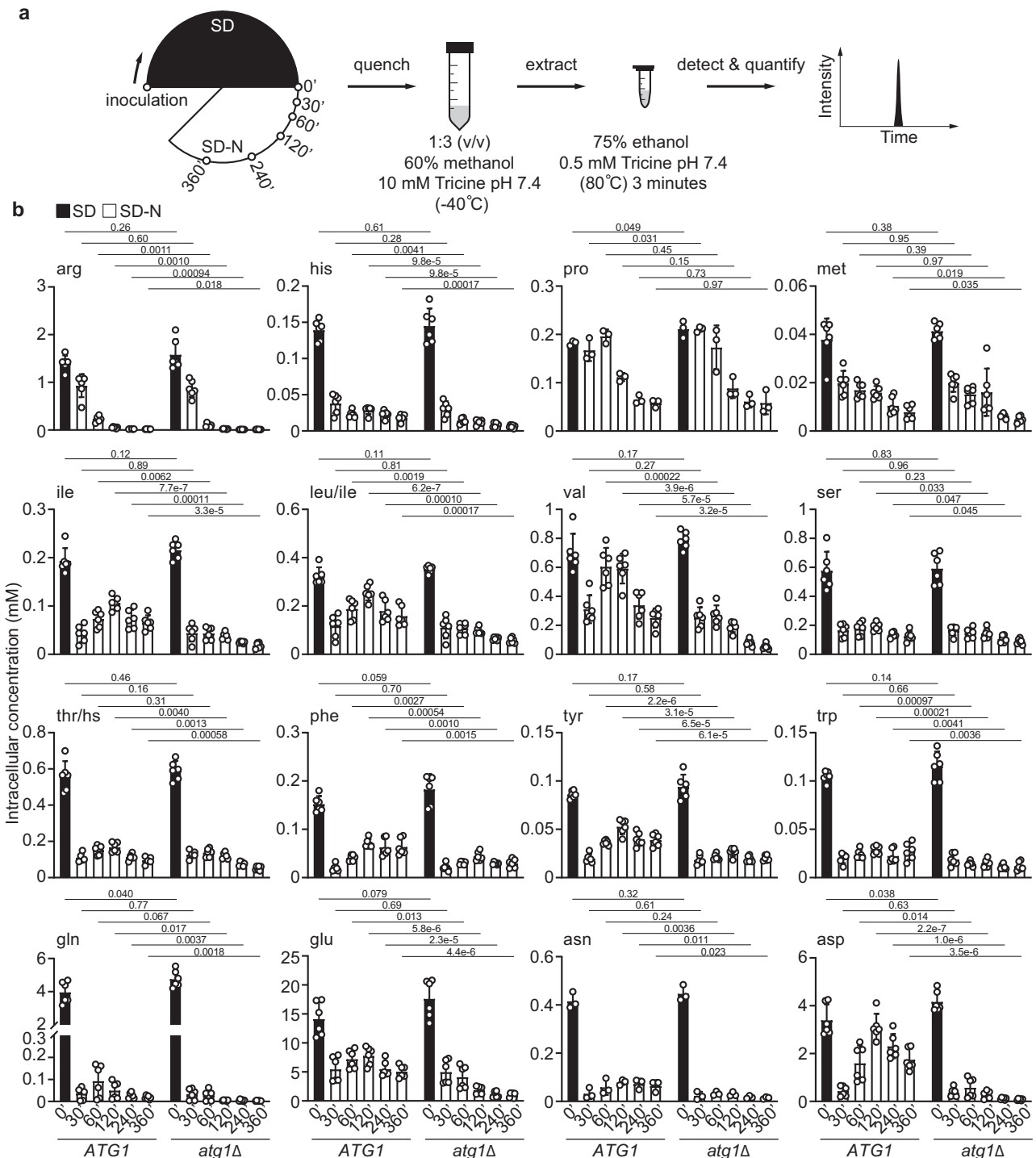

**Fig. 1 Autophagy is crucial for maintaining amino acid pools under nitrogen starvation. a** Schematic of the experimental design. WT (*ATG1*) and *atg1*Δ cells were grown in synthetic defined medium (SD) and starved for nitrogen (SD-N) for varying times. Metabolites were extracted and analyzed by LC-MS/MS. **b** Changes of amino acids in nitrogen-starved WT and *atg1*Δ cells. Three independent replicates for pro and asn, and six for the remaining amino acids. Absolute concentration was calculated based on Supplementary Fig. 3b by comparing different time points of WT and *atg1*Δ cells to WT grown in SD. WT and *atg1*Δ cells were always processed and analyzed together and therefore their abundances are directly comparable. hs: homoserine. *P* values were calculated using unpaired two-sided Student's *t* test assuming equal variances. Data are presented as mean ± standard deviation.

aspartate and glutamate that their uniformly labeled (i.e., [U-$^{13}$C]) species are the dominant isotopologue under nitrogen starvation, although their partially labeled species are also present to varying extent (Supplementary Figs. 6a–6c).

When cells were switched from SD to [U-$^{13}$C]-SD-N, we could detect many [U-$^{13}$C]-amino acids (Fig. 2b and Supplementary

Fig. 5c), suggesting that nitrogen-starved cells are still capable of synthesizing amino acids. However, a careful examination of these [U-$^{13}$C]-amino acids revealed two peculiar disparities. First, cells appeared to synthesize amino acids in varying capacity under nitrogen starvation, ranging from minimal (e.g., histidine and arginine) to substantial (e.g., aspartate and glutamate) (Fig. 2b

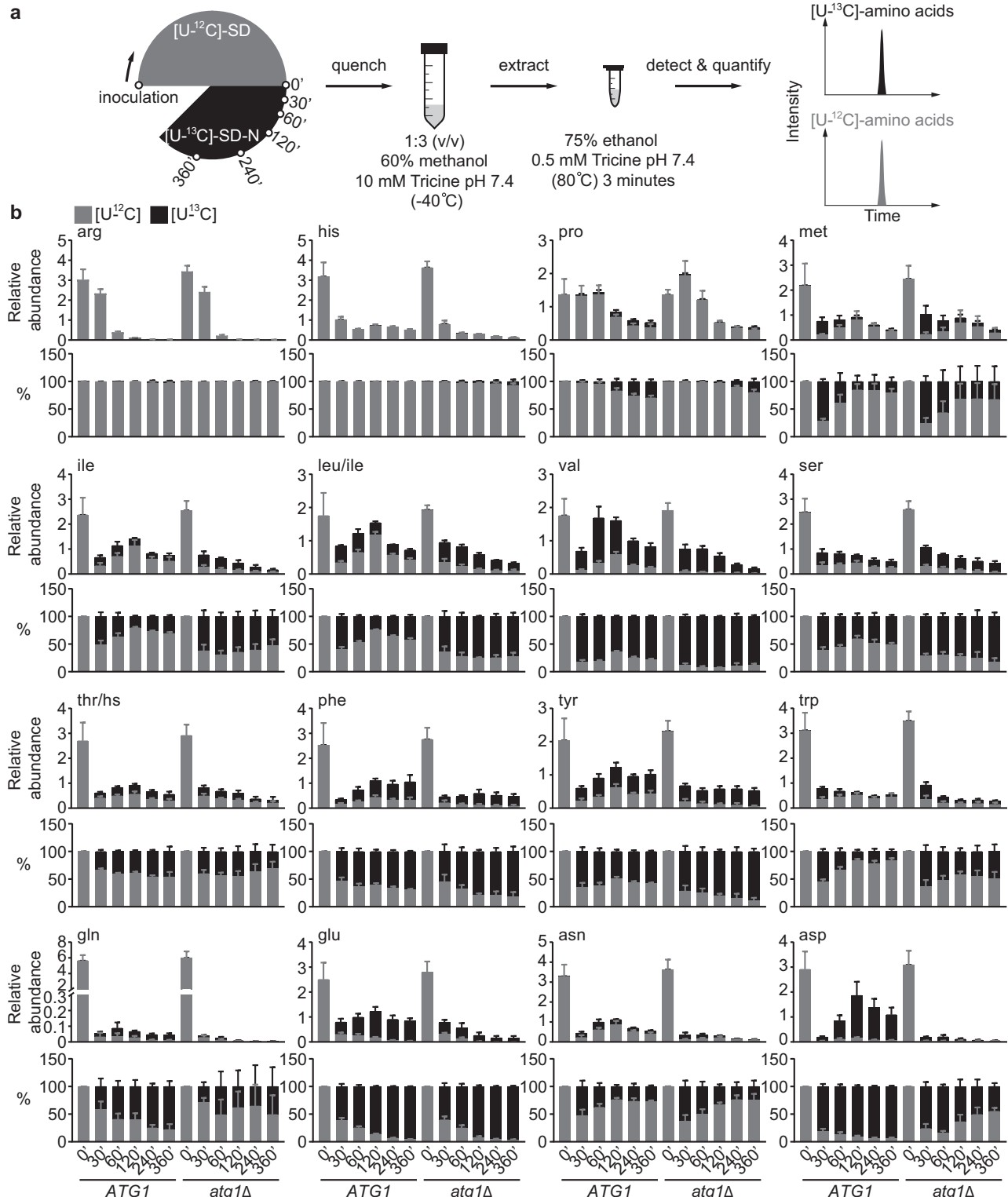

**Fig. 2 Autophagy promotes the synthesis of specific amino acids under nitrogen starvation. a** Schematic of the experimental design. WT (*ATG1*) and *atg1Δ* cells were grown in SD and starved in [U-13C]-SD-N for varying times. Metabolites were extracted and analyzed by LC-MS/MS. **b** Changes of [U-12C]- and [U-13C]-amino acids in nitrogen-starved WT and *atg1Δ* cells. Three independent replicates for leu/ile, five for his, nine for asp, and six for the remaining amino acids. WT and *atg1Δ* cells were always processed and analyzed together and therefore their relative abundances are directly comparable. hs: homoserine.

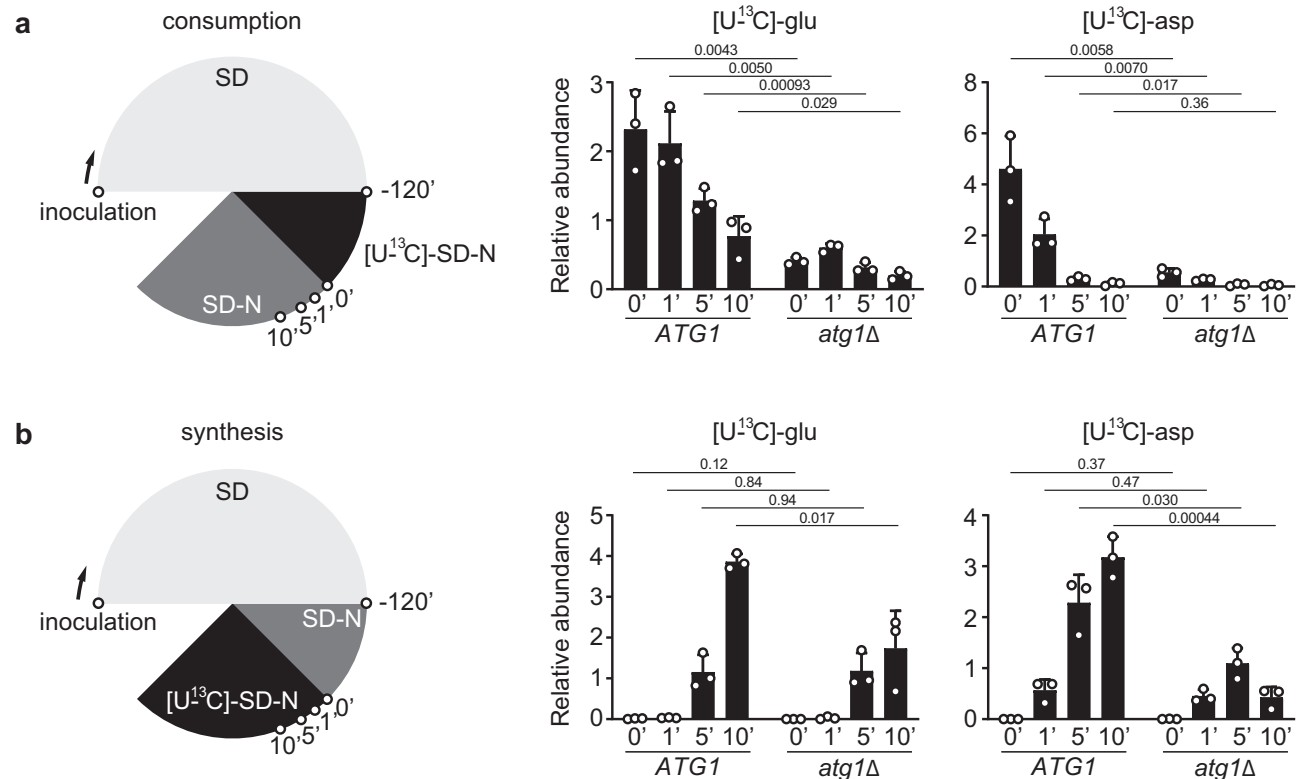

**Fig. 3 Autophagy promotes glutamate and aspartate synthesis under nitrogen starvation. a** Glutamate and aspartate are consumed at comparable rates in WT (*ATG1*) and *atg1Δ* cells under nitrogen starvation. Three independent replicates. **b** Glutamate and aspartate accumulate more rapidly in WT cells under nitrogen starvation. Three independent replicates. *P* values were calculated using unpaired two-sided Student's *t* test assuming equal variances. Data are presented as mean ± standard deviation.

and Supplementary Fig. 5c). Second, the newly synthesized ([U-13C]) amino acids exhibited drastically different dependencies on autophagy. For instance, autophagy appeared to be totally dispensable for amassing [U-13C]-serine and [U-13C]-tyrosine, whereas accumulation of [U-13C]-aspartate and [U-13C]-glutamate was absolutely dependent on autophagy (Fig. 2b and Supplementary Fig. 5c). Interestingly, the onset of autophagy between 60-120 minutes under nitrogen starvation was accompanied by a gradual decline in the [U-13C]-fraction of several amino acids (e.g., tryptophan, leucine, isoleucine, and valine) in WT cells (Fig. 2b and Supplementary Fig. 5c), consistent with the notion that autophagy degrades proteins to supply amino acids. Importantly, we found that the strong accumulation of [U-13C]-aspartate and [U-13C]-glutamate in nitrogen-starved WT cells was due to enhanced synthesis, rather than reduced consumption (Figs. 3a and 3b). Taken together, these results suggest that autophagy promotes the synthesis of specific amino acids such as aspartate and glutamate under nitrogen starvation.

Notably, in the presence of nitrogen (i.e., in SD medium) where autophagy operates at basal levels, ammonium and glucose assimilation into amino acids was virtually indistinguishable between WT and *atg1Δ* cells (Supplementary Figs. 7a–7d). Inducing autophagy by rapamycin treatment[30] (Fig. 4a) did not elicit the changes of glutamate and aspartate as observed under nitrogen starvation (Fig. 4b). Perhaps this is not surprising, given that autophagy, generally considered to be beneficial, actually conferred a growth disadvantage following rapamycin treatment (Fig. 4c). Moreover, glutamine supplementation rescued glutamate and aspartate levels in nitrogen-starved *atg1Δ* cells compared to WT cells (Supplementary Fig. 8). A stable isotope-tracing experiment revealed that glutamine promoted glutamate synthesis primarily through deamination, with glutamate

subsequently transferring its amino group to aspartate (Supplementary Fig. 8). Collectively, these observations reinforce the notion that autophagy functions primarily as a nutrient-starvation response and further suggest that the metabolic response to sustain glutamate and aspartate synthesis depends on both autophagy and the nutrient status (i.e., nitrogen starvation).

**Autophagy provides ammonium for the GS-GOGAT pathway to synthesize glutamate**. How does autophagy promote glutamate and aspartate synthesis under nitrogen starvation? Addressing this question requires a thorough examination of two key nutrients for amino acid synthesis: carbon and nitrogen. Using [U-14C]-glucose, we found that *atg1Δ* cells transported slightly less glucose than WT cells under nitrogen starvation (~30%, Supplementary Figs. 9a and 9b). This uptake defect, however, is unlikely to explain the diminished glutamate and aspartate synthesis in *atg1Δ* cells, because of the comparable levels of [U-13C]-glycolytic and [U-13C]-TCA intermediates between WT and *atg1Δ* cells (Supplementary Fig. 5a). In fact, levels of α-ketoglutarate, the immediate precursor for glutamate synthesis, were even higher in nitrogen-starved *atg1Δ* cells (Supplementary Fig. 5a). With respect to aspartate, we were unable to reliably measure oxaloacetate (the immediate precursor for aspartate) due to its labile nature[31]. However, the modestly higher levels of [U-13C]-citrate (immediately downstream of oxaloacetate) in nitrogen-starved *atg1Δ* cells (Supplementary Fig. 5a) would argue against oxaloacetate as limiting for aspartate synthesis. Therefore, we conclude that autophagy allows nitrogen-starved yeast cells to primarily retrieve nitrogen to sustain glutamate and aspartate synthesis.

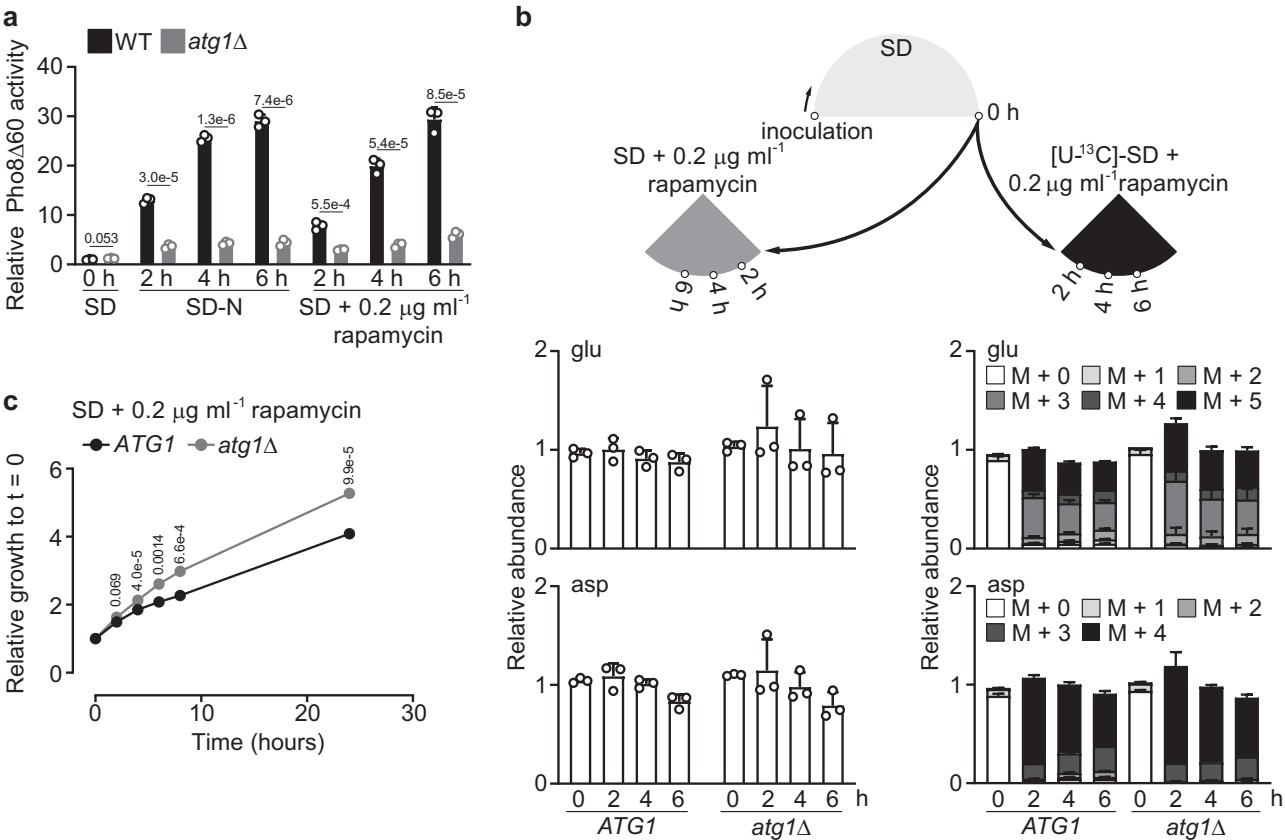

**Fig. 4 Characterization of rapamycin-induced autophagy in the presence of nitrogen. a** Rapamycin treatment induces autophagy in the presence of nitrogen. Three independent replicates. All samples were normalized to WT cells grown in SD. **b** Changes of glutamate and aspartate in rapamycin-treated WT (*ATG1*) and *atg1Δ* cells in the presence of nitrogen. Three independent replicates. **c** Autophagy deficiency renders yeast cells more recalcitrant to growth inhibition by rapamycin. Cells were grown in SD to log phase, and washed and resuspended in prewarmed SD + rapamycin with a starting $OD_{600}$ ~0.1. Growth was monitored by measuring $OD_{600}$ at indicated time intervals. Three independent replicates. *P* values were calculated using unpaired two-sided Student's *t* test assuming equal variances. Data are presented as mean ± standard deviation.

Amassing glutamate, the major (~80%) cellular nitrogen donor[25], would effectively replenish cellular nitrogen currency and mitigate the perils of nitrogen starvation. But how might autophagy enable cells to retrieve nitrogen without an external nitrogen supply? Yeast cells host a complex metabolic network that allows them to utilize diverse nitrogen sources to support glutamate synthesis. With ammonium, the glutamate dehydrogenase (GDH) and glutamine synthetase coupled to glutamine oxoglutarate aminotransferase (or glutamate synthase) (GS-GOGAT) pathways are responsible for assimilating this inorganic nitrogen into glutamate[32] (Fig. 5a). With organic nitrogenous compounds, yeast cells employ an intricate network of deamination and transamination reactions to support glutamate synthesis[33]. Deamination liberates ammonium that is subsequently assimilated by the aforementioned GDH and GS-GOGAT pathways and there exist a plethora of amino acid deaminases for such purposes, e.g., Gdh2p for glutamate[34], Cha1p for threonine and serine[35], Asp1p for asparagine[36], and Ilv1p for threonine[37]. Moreover, yeast cells could also break down arginine to liberate ammonium via the combined action of arginase and urea amidolyase[38,39]. On the other hand, transamination allows for the direct transfer of the amino group from certain nitrogenous compounds (e.g., amino acids) to α-ketoglutarate to synthesize glutamate. For instance, the Ehrlich pathway is a well-known route for transaminating aromatic and branched-chain amino acids[40]. However, genetically dissecting the contribution of deamination and transamination reactions is challenging because of the overwhelming number of enzymes that are potentially involved.

Such complexity led us to first examine whether ammonium was a relevant nitrogen source that autophagy might supply under nitrogen starvation. To this end, we disrupted *GDH1*, *GDH3*, and *GLT1* to inactivate both the GDH and GS-GOGAT pathways. When cells were shifted from glutamate-fortified SD to [U-13C]-SD-N (Fig. 5b), loss of ammonium assimilation reduced [U-13C]-glutamate and [U-13C]-aspartate by ~50% (Fig. 5c), suggesting that ammonium is an important nitrogen source for glutamate and aspartate synthesis in nitrogen-starved cells. The remaining [U-13C]-glutamate in the *gdh1Δgdh3Δglt1Δ* mutant implies that transamination also contributes substantially to glutamate synthesis under nitrogen starvation.

To examine the contribution of each pathway to ammonium assimilation under nitrogen starvation, we created two mutants, *gdh1Δgdh3Δ* and *glt1Δ*, to inactivate the GDH or GS-GOGAT pathway individually. Strikingly, we found that only the GS-GOGAT pathway was relevant for assimilating ammonium under nitrogen starvation, as loss of the GDH pathway showed little impact on glutamate and aspartate synthesis (Fig. 5d). This preference may likely be due to the relatively low affinity of Gdh1p and Gdh3p for ammonium ($K_m$ values ~6 mM and 5 mM, respectively)[41]. Nitrogen starvation may lower ammonium concentration well below their $K_m$, thus rendering them virtually inoperative. Importantly, *glt1Δ* cells accumulated significantly more glutamine than WT cells in an autophagy-dependent fashion, most of which was fully labeled with 13C (Fig. 5e). To identify the ammonium source, we next performed an isotope tracing experiment where cells were labeled with 15N

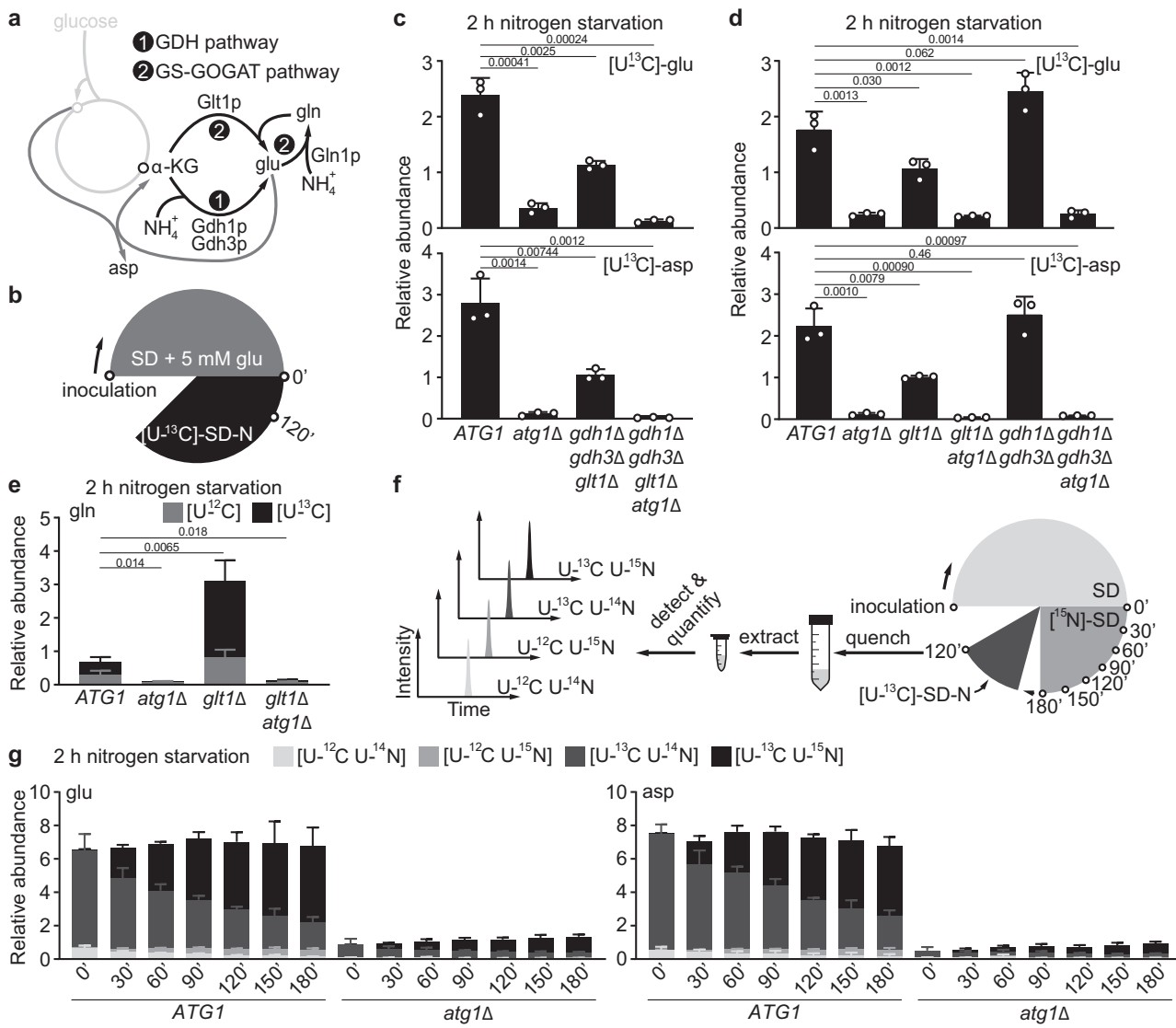

**Fig. 5 The GS-GOGAT pathway assimilates ammonium to synthesize glutamate under nitrogen starvation. a** Schematic of ammonium assimilation in *S. cerevisiae*. GDH: glutamate dehydrogenase (Gdh1p and Gdh3p); GS-GOGAT: glutamine synthetase (Gln1p)-glutamine oxoglutarate aminotransferase (Glt1p); α-KG: α-ketoglutarate. There are two major routes for glucose-derived carbon to enter the TCA cycle: acetyl-CoA to citrate (catalyzed by Cit1p and Cit3p) and pyruvate to oxaloacetate (catalyzed by Pyc1p and Pyc2p). **b** Schematic of the experimental design for **c**, **d**, and **e**. Cells were grown in glutamate-fortified SD medium and shifted to [U-$^{13}$C]-SD-N for 2 h. **c** Loss of ammonium assimilation impairs glutamate synthesis under nitrogen starvation. Three independent replicates. **d** Glutamate synthesis is impeded in *glt1Δ* cells, but not in *gdh1Δgdh3Δ* cells under nitrogen starvation. Three independent replicates. **e** The GS-GOGAT pathway predominantly assimilates ammonium to support glutamate synthesis under nitrogen starvation. Three independent replicates. **f** Schematic of experimental design for **g**. Cells grown in SD were labeled with $^{15}$N for varying times and then starved in [U-$^{13}$C]-SD-N for 2 h. **g** Autophagy degrades macromolecules (e.g., proteins) to provide nitrogen for glutamate and aspartate synthesis under nitrogen starvation. Three independent replicates. *P* values were calculated using unpaired two-sided Student's *t* test assuming equal variances. Data are presented as mean ± standard deviation.

ammonium for varying times and shifted to [U-$^{13}$C]-SD-N for 2 h (Fig. 5f and Supplementary Fig. 10a). Our rationale is based on the fact that incorporation of $^{15}$N into nitrogenous compounds follows distinct dynamics: shorter times would favor labeling (or exchange) of ammonium and small nitrogenous compounds (e.g., amino acids), but not macromolecules (e.g., proteins). In SD medium containing nitrogen, a short pulse such as 30 minutes was sufficient to label most amino acids predominantly with $^{15}$N (Supplementary Figs. 7c and 10b). However, following nitrogen starvation, $^{14}$N nitrogen was persistent in the newly synthesized (i.e., [U-$^{13}$C]) glutamate, aspartate, and glutamine (Fig. 5g and Supplementary Fig. 10c), indicating that macromolecules (e.g., proteins) are a significant ammonium source provided by

autophagy under nitrogen starvation. Moreover, when fed with low amounts of $^{15}$N ammonium (10 μM) under nitrogen starvation, WT cells incorporated more $^{15}$N into glutamate and aspartate (Supplementary Fig. 10d). These observations, together with the similar protein levels of the relevant enzymes between WT and *atg1Δ* cells (Supplementary Fig. 10e), suggest that autophagy degrades macromolecules (e.g., proteins) under nitrogen starvation to provide ammonium, which is primarily assimilated by the GS-GOGAT pathway to synthesize glutamine, and subsequently glutamate and aspartate.

**Aspartate synthesis is a major nitrogen investment under nitrogen starvation.** Amassing glutamate enables nitrogen-starved cells

to reinvest in nitrogen anabolism, and yet they do not appear to ration out glutamate evenly. This interpretation was initially inferred from the peculiar disparity in the contribution of autophagy to the synthesis of [U-13C]-amino acids (Fig. 2b and Supplementary Fig. 5c). For example, aspartate, phenylalanine, tyrosine, valine, isoleucine, and leucine all acquire their amino group from glutamate (directly or indirectly), but aspartate synthesis and to a lesser extent valine synthesis was heavily dependent on autophagy (Fig. 2b and Supplementary Fig. 5c). One possible explanation is that nitrogen-starved cells invest significantly more glutamate in aspartate synthesis than many other amino acids. As a result, the limited glutamate in atg1Δ cells only minimally impacted synthesis of phenylalanine, tyrosine, leucine, and isoleucine, but severely restricted aspartate synthesis under nitrogen starvation.

To test this hypothesis, we disrupted AAT1 and AAT2 that encode the two (reversible) aspartate aminotransferases[42,43] (Fig. 6a). Single and double mutants were grown in aspartate-fortified SD and shifted to [13C]-SD-N for 2 h. Consistent with our earlier observations (Fig. 2b and Supplementary Fig. 5c), newly synthesized aspartate and glutamate were strongly dependent on autophagy (Fig. 6b). Loss of Aat1p reduced [U-13C]-aspartate modestly, whereas synthesis of [U-13C]-aspartate was almost completely abolished in aat2Δ cells (Fig. 6b), suggesting that Aat2p is the major enzyme for aspartate synthesis under nitrogen starvation. Interestingly, Aat2p is reportedly a cytosolic enzyme[44], implying that aspartate synthesis occurs mainly in the cytoplasm under nitrogen starvation. Importantly, aat2Δ cells accumulated approximately tenfold higher glutamate than WT cells, most of which was fully labeled with 13C (Fig. 6b). By contrast, loss of Aro8p and Aro9p, the two (reversible) aminotransferases for phenylalanine and tyrosine synthesis, failed to elicit the drastic changes of [U-13C]-glutamate as observed in aat2Δ cells (Fig. 6c). Moreover, abolishing valine and isoleucine synthesis as in the bat1Δbat2Δ mutant only led to modest accumulation of [U-13C]-glutamate (approximately twofold) under nitrogen starvation (Fig. 6d). Notably, the bat1Δbat2Δ mutant is reportedly auxotrophic for all three branched amino acids[45]. However, we found that it could still synthesize leucine (Fig. 6d) and therefore did not require leucine to grow (Supplementary Fig. 11). Nonetheless, these results collectively support our hypothesis that aspartate synthesis is a major nitrogen investment under nitrogen starvation. Lastly, in nitrogen-starved aat1Δaat2Δ, aro8Δaro9Δ, and bat1Δbat2Δ cells, we could readily observe an autophagy-dependent accumulation of the relevant, unlabeled [U-12C]-amino acids (Fig. 6b–d), which is consistent with our model where autophagy feeds transamination reactions to support glutamate synthesis under nitrogen starvation (Fig. 6a).

**Aspartate sustains protein and nucleic acid synthesis under nitrogen starvation.** Why would nitrogen-starved cells allocate a large amount of glutamate to aspartate synthesis? Aspartate is a precursor for several amino acids (Fig. 7a). The modestly elevated [U-13C]-threonine/homoserine and [U-13C]-asparagine in WT cells (Fig. 2b and Supplementary Fig. 5c) suggests a possible role for aspartate in promoting synthesis of its derivative amino acids. Aspartate also provides building blocks for proteins and nucleic acids (Fig. 7a). To test whether amassing aspartate via autophagy might support macromolecule synthesis under nitrogen starvation, we designed a radioisotope-tracing experiment to track how cells might utilize aspartate under nitrogen starvation (Fig. 7b).

Nitrogen-starved WT cells amassed significantly higher aspartate (Figs. 1b and 7c, and Supplementary Fig. 3a) and took up modestly more [U-14C]-aspartate (Fig. 7d). Importantly,

WT cells incorporated significantly more equivalents of aspartate (aspartate and its derivatives) into macromolecules (Fig. 7e, f). By selectively degrading nucleic acids (Supplementary Figs. 12a and 12b), we found that WT cells incorporated significantly more equivalents of aspartate into proteins (Fig. 7g, h), which is consistent with the notion that autophagy is important for sustaining protein synthesis under nitrogen starvation[22]. Given that aspartyl-tRNA was charged predominantly with newly synthesized aspartate under nitrogen starvation (Supplementary Figs. 13a and 13b), de novo aspartate synthesis is likely more important than pure recycling for translation. To further test the importance of de novo amino acid synthesis for translation under nitrogen starvation, we examined the glt1Δ mutant using 14C-radioistope tracing and found that it incorporated modestly less equivalents of valine (Supplementary Figs. 14a–14f) and aspartate (Supplementary Figs. 15a–15e) into proteins, suggesting that de novo glutamate and/or aspartate synthesis is important for supporting protein synthesis under nitrogen starvation.

On the other hand, measuring 14C in purified nucleic acids revealed that WT cells also assimilated more equivalents of aspartate into nucleic acids under nitrogen starvation (Fig. 7i, j). Similarly, impairing glutamate and aspartate synthesis (using the glt1Δ mutant) slightly but significantly reduced incorporation of aspartate equivalents into nucleic acids (Supplementary Figs. 15f and 15g), suggesting the importance of de novo glutamate and/or aspartate synthesis in supporting nucleic acid synthesis under nitrogen starvation. Subsequent nucleic acid fractionation revealed that aspartate-derived 14C was incorporated into DNA and mRNA, whereas incorporation into rRNA was limited (Supplementary Figs. 16a–16c). Interestingly, disrupting URA3 to block pyrimidine synthesis from aspartate only reduced aspartate-derived 14C incorporation into nucleic acids in the presence of nitrogen, but not under nitrogen starvation (Supplementary Figs. 17a–17g). One possible explanation is compensation by alternative pathways (Supplementary Fig. 17b). Lastly, like atg1Δ cells, the autophagy-deficient atg8Δ and pep4Δprb1Δ cells also showed significantly diminished incorporation of aspartate equivalents into macromolecules (Fig. 8a–i). Collectively, these results suggest that a major function of aspartate synthesis enabled by autophagy is to sustain macromolecule synthesis (i.e., protein and nucleic acids) under nitrogen starvation.

## Discussion

Autophagy empowers eukaryotic cells to cope with nutrient starvation by catabolizing cellular constituents such as proteins. However, yeast cells do not simply reutilize the derived amino acids as often assumed[16,18,22,24]; instead, they feed these amino acids into an intricate network of transamination and deamination reactions to support glutamate synthesis (Fig. 9). This complex metabolic response allows nitrogen-starved cells to effectively replenish their nitrogen currency and to reinvest in nitrogen anabolism such as aspartate synthesis to sustain macromolecule synthesis. Our work might provide a mechanistic explanation for how autophagy enables nitrogen-starved yeast cells to finish their cell cycle[46,47], and it might also shed light on the intimate link between autophagy deficiency and genome instability observed in yeast and mammalian cells[46,48–50]. These notions are in line with a previous study by Guo et al.[6], who demonstrated that autophagy allows cancer cells to recycle nucleosides to maintain nucleotide pool homeostasis and to promote survival under nutrient starvation.

In light of our work from yeast, it is quite tantalizing to ask: would autophagy facilitate a similar metabolic rewiring in nutrient-

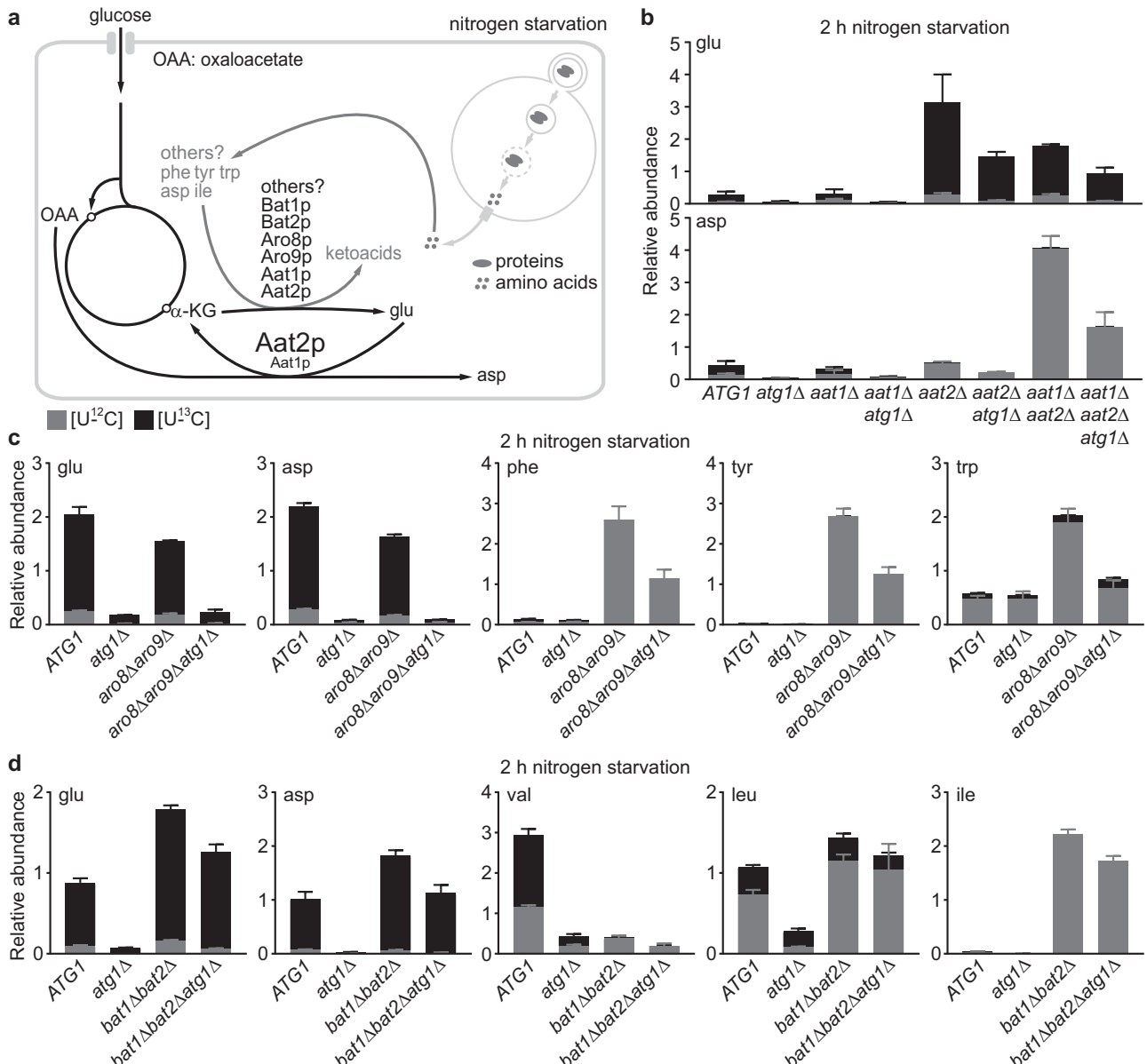

**Fig. 6 Aspartate is a major nitrogen investment under nitrogen starvation. a** Proposed model for glutamate and aspartate metabolism under nitrogen starvation. Only ammonium-independent glutamate synthesis is shown. The large font for Aat2p indicates that it acts as the major enzyme for aspartate synthesis under nitrogen starvation. **b** Loss of aspartate synthesis leads to strong accumulation of newly synthesized glutamate. Cells were grown in SD + 10 mM aspartate (pH 5.4) and starved in [U-13C]-SD-N for 2 h. Two independent replicates for *aat2Δatg1Δ* (aspartate), five for *aat1Δaat2Δatg1Δ*, and three for the remaining strains. **c** Loss of phenylalanine and tyrosine synthesis does not significantly affect newly synthesized glutamate under nitrogen starvation. Cells were grown in SD + phenylalanine, tyrosine, and tryptophan at 1 mM each and shifted to [U-13C]-SD-N for 2 h. The *aro8Δaro9Δ* mutant is not auxotrophic for tryptophan, as evidenced by the detection of [U-13C]-tryptophan under nitrogen starvation. Nonetheless, we supplemented tryptophan to compare it with phenylalanine and tyrosine as an amino-group donor for glutamate synthesis under nitrogen starvation. Three independent replicates. **d** Loss of valine and isoleucine synthesis led to modest accumulation of newly synthesized glutamate under nitrogen starvation. Strains were grown in SD + isoleucine, leucine, and valine at 1 mM each and shifted to [U-13C]-SD-N for 2 h. Three independent replicates. The presence of [U-13C]-leucine in the *bat1Δbat2Δ* mutant indicates the existence of additional enzyme(s) for leucine synthesis. Data are presented as mean ± standard deviation.

starved mammalian cells? In mice, autophagy is crucial for maintaining amino acid pools under nutrient deprivation[9,11,13]. Moreover, mammalian cells have the necessary metabolic machinery to perform similar metabolic reactions and breast cancer cells can recycle ammonium, generally regarded as a toxic waste product, to synthesize glutamate and support nitrogen anabolism[51]. Given that certain amino acids are particularly important for growth and proliferation, e.g., glutamine and aspartate[52–57], amassing these amino acids via autophagy would be conceivably beneficial for surviving starvation. In the case of cancer, such a metabolic

response could potentially allow cancer cells to adapt more effectively to and survive their harsh microenvironment or to cope with DNA-damaging chemotherapies, an interesting and important scenario that warrants further investigation.

## Methods
**Strains and culture conditions**. Prototrophic CEN.PK *Saccharomyces cerevisiae*[58] was used for strain construction (see Supplementary Table 1). Lithium-acetate-based transformation was performed to replace genes of interest with drug resistant markers[59]. Unless otherwise stated, yeast strains were grown in SD (20 g l⁻¹

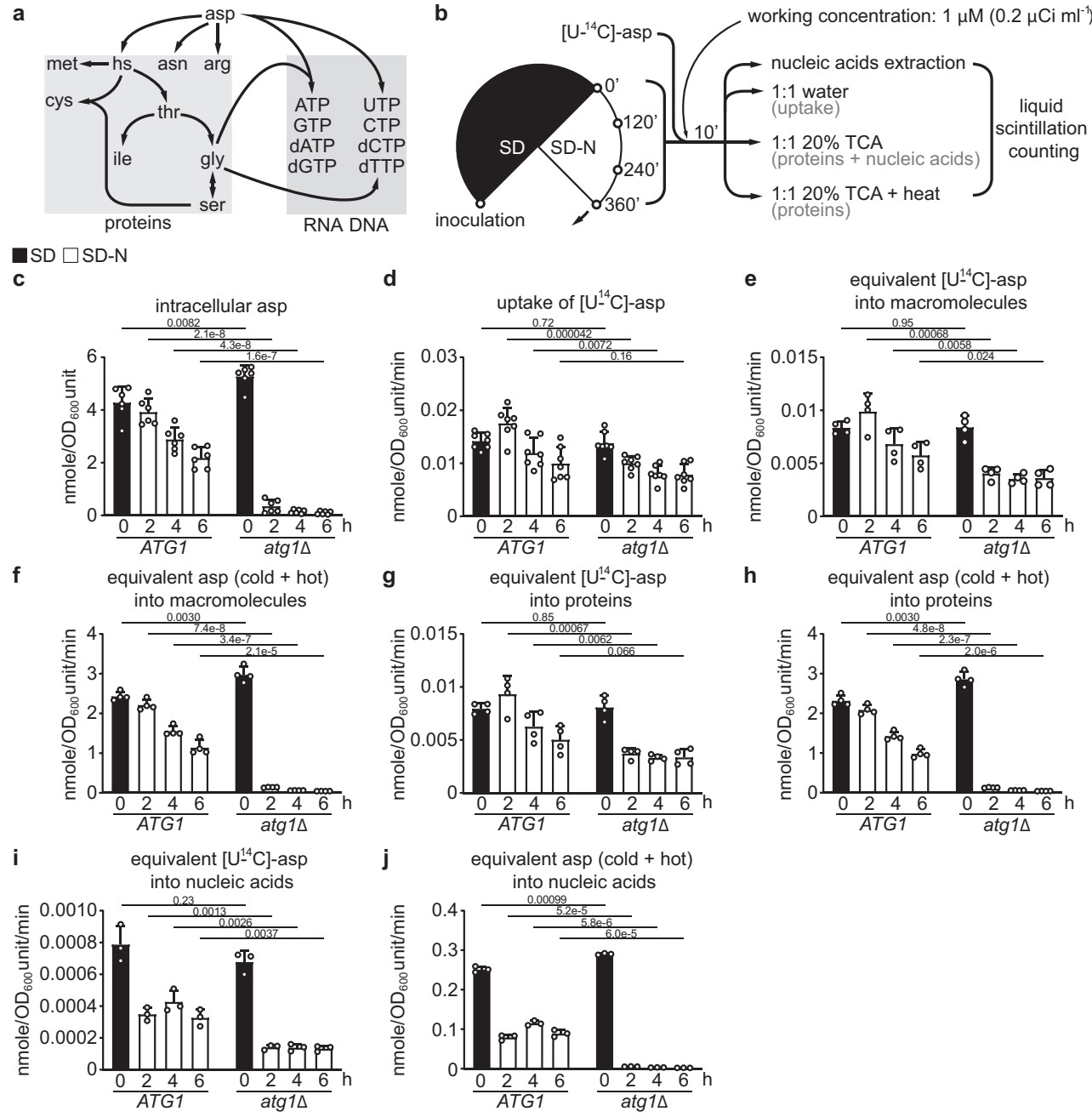

**Fig. 7 Aspartate sustains macromolecule synthesis under nitrogen starvation. a** Metabolic fate of aspartate in cellular anabolism. hs: homoserine. **b** Schematic of the experimental design for **d**, **e**, **g**, and **i**. **c** Estimation of intracellular aspartate concentration in WT (*ATG1*) and *atg1*Δ cells. Data were based on Fig. 1b and Supplementary Fig. 3b. Six independent replicates. **d** Uptake of [U-14C]-aspartate in WT and *atg1*Δ cells. Seven independent replicates. **e** Incorporation of equivalents of [U-14C]-aspartate (aspartate and its derivatives) into macromolecules (proteins and nucleic acids). Four independent replicates. **f** Incorporation of total aspartate equivalents into macromolecules. Four independent replicates. **g** Incorporation of equivalents of [U-14C]-aspartate into proteins. Four independent replicates. **h** Incorporation of total aspartate equivalents into proteins. Four independent replicates. **i** Incorporation of equivalents of [U-14C]-aspartate into nucleic acids. Three independent replicates. **j** Incorporation of total aspartate equivalents into nucleic acids. Three independent replicates. *P* values were calculated using unpaired two-sided Student's *t* test assuming equal variances. Data are presented as mean ± standard deviation.

glucose and 6.7 g l⁻¹ BD Difco yeast nitrogen base without amino acids) or starved in SD-N medium (20 g l⁻¹ glucose and 1.7 g l⁻¹ BD Difco yeast nitrogen base without amino acids and ammonium sulfate) at 30 °C and 300 rpm.

**Metabolite extraction and detection.** The extraction protocol comprised two sequential steps: quenching and extraction, which were originally published by Castrillo et al.[60] and Gonzalez et al.[61], respectively. Quenching was achieved by mixing one volume of cell culture with three volumes of methanol–water solution (60% v/v, buffered with 10 mM Tricine to pH 7.4) kept at −40 °C. Quenched cells

were centrifuged and resuspended in extraction buffer containing ethanol-water (75% v/v, buffered with 0.5 mM Tricine to pH 7.4) and heated at 80 °C for three minutes. Cell extracts were immediately chilled on ice and subsequently centrifuged at maximum speed at 0 °C to remove cell debris. The supernatant was vacuum-dried and stored at −80 °C until analysis.

Samples were analyzed using reversed-phase and normal-phase HPLC coupled to tandem mass spectrometry as described[62,63]. In the reversed-phase method, metabolites were separated on a Synergi Fusion-RP column (4 µm particle size, 80 Å pore size, 150 mm × 2 mm, Phenomenex) using a Shimadzu Prominence HPLC machine and simultaneously detected by a triple quadrupole mass

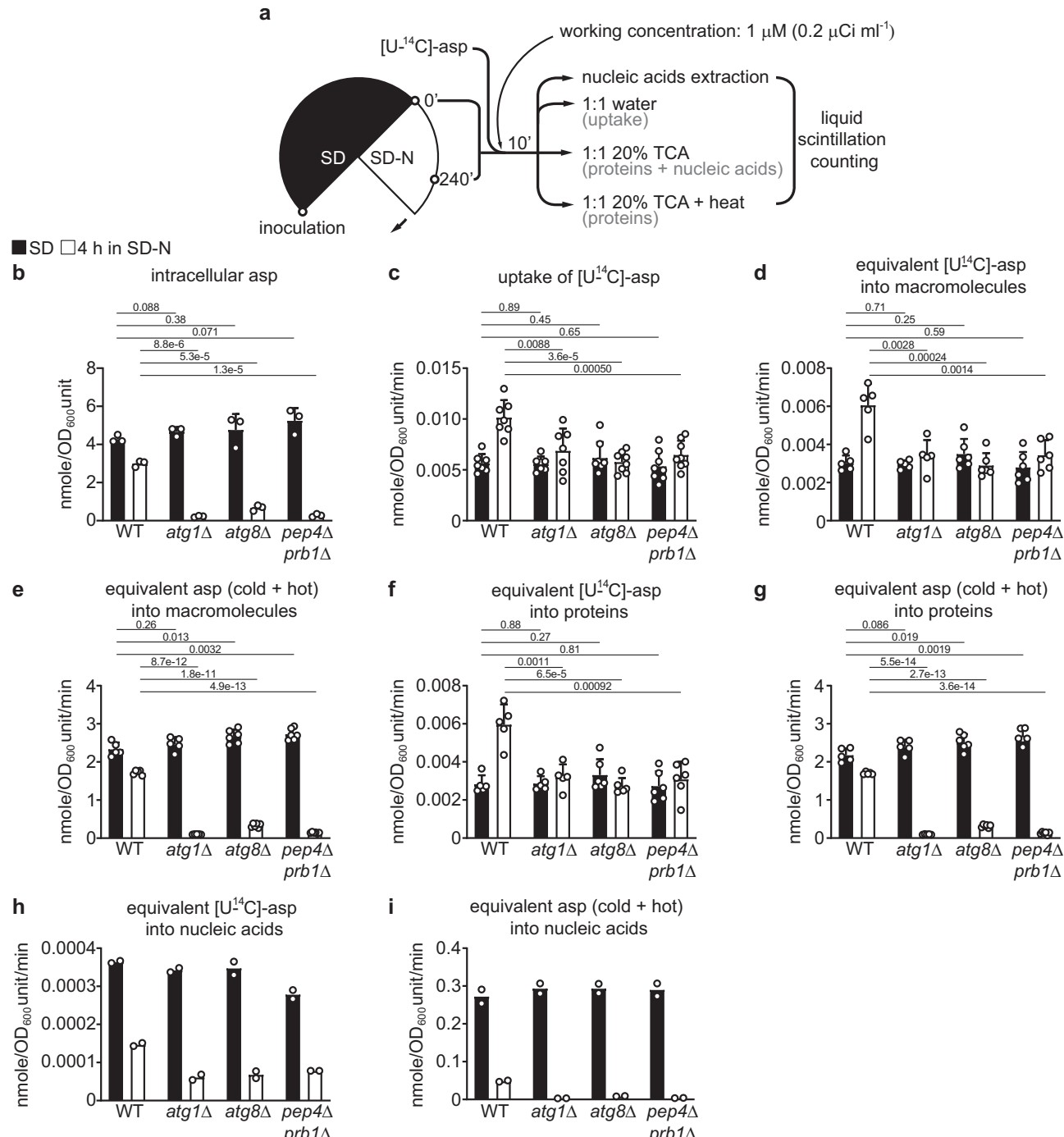

**Fig. 8 Other autophagy-deficient mutants are also defective in assimilating aspartate into macromolecules under nitrogen starvation. a** Schematic of the experimental design. **b** Estimation of intracellular aspartate concentration. Three independent replicates. **c** Uptake of [U-14C]-aspartate. Seven independent replicates for WT and atg1Δ, and eight for atg8Δ and pep4Δprb1Δ. **d** Incorporation of equivalents of [U-14C]-aspartate (aspartate and its derivatives) into macromolecules. Five independent replicates for WT and atg1Δ, and six for atg8Δ and pep4Δprb1Δ. **e** Incorporation of total aspartate equivalents into macromolecules. Five independent replicates for WT and atg1Δ, and six for atg8Δ and pep4Δprb1Δ. **f** Incorporation of equivalents of [U-14C]-aspartate into proteins. Five independent replicates for WT and atg1Δ, and six for atg8Δ and pep4Δprb1Δ. **g** Incorporation of total aspartate equivalents into proteins. Five independent replicates for WT and atg1Δ, and six for atg8Δ and pep4Δprb1Δ. **h** Incorporation of equivalents of [U-14C]-aspartate into nucleic acids. Two independent replicates. **i** Incorporation of total aspartate equivalents into nucleic acids. Two independent replicates. P values were calculated using unpaired two-sided Student's t test assuming equal variances. Data are presented as mean ± standard deviation for **b**–**g** and mean only for **h** and **i**.

spectrometer (3200 QTRAP, AB SCIEX). The total run time was 22 minutes at a flow rate of 0.5 ml min$^{-1}$. Three methods that differed by additives were employed: (1) 0.1% (v/v) formic acid in water as Solvent A and 0.1% (v/v) formic acid in methanol as Solvent B; (2) 5 mM ammonium acetate (pH 5.5) in water as Solvent A and 5 mM ammonium acetate in methanol as Solvent B; (3) 10 mM tributylamine

(pH 5.0) in water as Solvent A and methanol as Solvent B. The following gradient elution was performed: 0.01 min, 0% B, 4 min, 0% B, 11 min, 50% B, 13 min, 100% B, 17 min, 100% B, 18 min, 0% B, 22 min, 0% B. Metabolites were detected by multiple reaction monitoring (MRM) with transitions listed in Supplementary Table 2. In the normal-phase method, metabolites were separated on a SeQuant®

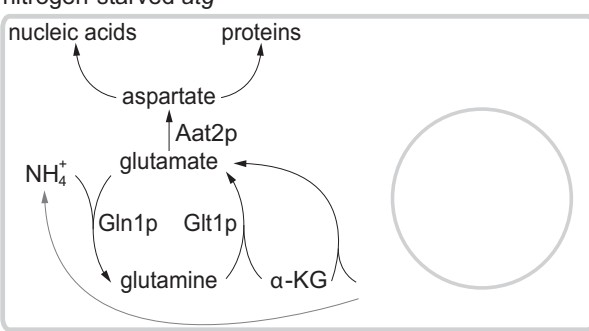

**Fig. 9 Model.** Under nitrogen starvation, autophagy allows yeast cells to retrieve nitrogen to support glutamate synthesis via a complex nexus of transamination and deamination reactions. This replenishes cellular nitrogen currency for subsequent nitrogen anabolism such as aspartate synthesis to support macromolecule synthesis. Although not specifically depicted here, amino acids derived directly from protein turnover may also participate in macromolecule synthesis under nitrogen starvation.

ZIC®-pHILIC column (5 μm particle size, 150 mm × 2.1 mm, Millipore Sigma) using a Shimadzu Prominence HPLC machine and simultaneously detected by a triple quadrupole mass spectrometer (Triple Quad 6500+, AB SCIEX). The total run time was 34 minutes at a flow rate of 0.15 ml min⁻¹, with 20 mM ammonium carbonate and 0.1% (v/v) ammonium hydroxide as Solvent A and acetonitrile as Solvent B. The following gradient was performed: 0.01 min 80% B, 20 min 20% B, 20.5 min 80% B, 34 min 80% B. Metabolites were detected by MRM with transitions listed in Supplementary Table 3.

Metabolites were quantified using Analyst® 1.6.2 and 1.6.3 packages (AB SCIEX) by calculating total peak area. Peak areas were first normalized to $OD_{600}$, which were further normalized to the average of all samples. For instance, normalization of peak areas by $OD_{600}$ gives a list of data for aspartate for WT and $atg1\Delta$ cells in SD and SD-N: $Asp_{SD}^{WT}$, $Asp_{SD-N}^{WT}$, $Asp_{SD}^{atg1\Delta}$, and $Asp_{SD-N}^{atg1\Delta}$. The relative abundance for $Asp_{SD}^{WT}$ is calculated as $Asp_{SD}^{WT}$/average($Asp_{SD}^{WT}$, $Asp_{SD-N}^{WT}$, $Asp_{SD}^{atg1\Delta}$, and $Asp_{SD-N}^{atg1\Delta}$). For the [U-¹³C]-tracing experiments, data were processed similarly, except that [U-¹²C]- and [U-¹³C]-isotopologues of a metabolite were summed up to calculate the average of all samples. It should be noted that strains were always compared and analyzed simultaneously and therefore the relative abundance is directly comparable.

**[U-¹⁴C]-aspartate tracing.** [U-¹⁴C]-aspartate (PerkinElmer) was added to 2 ml culture at a final concentration of 0.2 μCi ml⁻¹ (equivalent to 1 μM aspartate) and labeling proceeded for 10 minutes. An aliquot of 450 μl culture was mixed with 450 μl ice-cold water to estimate uptake. Two additional 450 μl aliquots were mixed with 450 μl 20% ice-cold TCA and incubated on ice for 10 minutes. One mixture was heated at 95 °C for 10 minutes to solubilize nucleic acids and immediately chilled on ice for five minutes. Cell suspensions were collected on a Whatman GF/F filter (GE Healthcare) and washed once with either 10 ml ice-cold water or 10 ml 5% ice-cold TCA. Filters were dried overnight and counted in five ml Ultima Gold F (PerkinElmer) using a Hitachi liquid scintillation counter (AccuFlex LSC-8000).

To estimate ¹⁴C incorporation into nucleic acids, yeast culture was labeled with 0.2 μCi ml⁻¹ [U-¹⁴C]-aspartate (equivalent to 1 μM aspartate) for 10 minutes and cell pellets were snap-frozen in ethanol-dry-ice bath or liquid nitrogen, and stored at −80 °C. Frozen cell pellets were thawed on ice, washed once with ice-cold water, and resuspended in 500 μl extraction buffer (2% Triton X-100, 1% SDS, 100 mM NaCl, 10 mM Tris-HCl pH 8.0, 1 mM EDTA) and 500 μl phenol (pH 8.0) (Fisher Scientific), and lysed with glass beads on a bead beater. The supernatant was extracted again with 500 μl phenol (pH 8.0), followed by a third extraction with 500 μl chloroform to remove the residual phenol. Nucleic acids were ethanol precipitated, washed with 70% ethanol, and resuspended in nuclease-free water. Absorbance at 260 nm was measured to estimate nucleic acid concentration. The ¹⁴C radioactivity from nucleic acids was counted in five ml Ultima Gold (PerkinElmer) using a Hitachi liquid scintillation counter (AccuFlex LSC-8000).

Nucleic acid fractionation was performed as described below. [U-¹⁴C]-aspartate-labeled cells were resuspended in 500 μl TES (10 mM Tris-HCl pH 7.5, 10 mM EDTA pH 8.0, and 0.5% SDS) and 500 μl acidic phenol (pH 4.3) (Sigma). Cells were lysed using glass beads on a bead beater and the supernatant was extracted again with 500 μl acidic phenol (pH 4.3), followed by a third extraction with chloroform to remove the residual phenol. RNA was ethanol precipitated, washed with 70% ethanol, and resuspended in nuclease-free water. mRNA was purified from total RNA using the Dynabeads mRNA purification kit (ThermoFisher Scientific) following the manufacturer's instructions. RNA in the flowthrough from mRNA purification was ethanol precipitated, washed with 70% ethanol, and resuspended in RNA loading solution (95% formamide, 0.02% SDS,

0.02% bromophenol blue, and 1 mM EDTA pH 8.0). Samples were heated at 75 °C for five minutes followed by rapid chilling on ice. Denatured RNA was loaded onto a 1.3% TAE low melting agarose gel. rRNAs (18S and 25S) were visualized by ethidium bromide staining, excised using a clean scalpel, and purified using the NuceloSpin Gel and PCR clean-up kit (Takara) following the manufacturer's instructions. DNA was purified from total nucleic acid by a combination of RNase A treatment and column purification. Briefly, total nucleic acid was digested with 25 μg RNase A (Epicenter) at 37 °C for 1 h and DNA was purified from the mixture using the NuceloSpin Gel and PCR clean-up kit (Takara) following the manufacturer's instructions.

**Estimation of absolute amounts of incorporated radioisotopes and total amino acids.** Sample CPM (counts per minute) was first converted to DPM (disintegrations per minute) using Eq. (1):

$$DPM = \frac{CPM}{Counting\ efficiency}. \tag{1}$$

Counting efficiency was estimated using radioisotope standards of known amounts in place of samples on filters or in solution. CPM was recorded and divided by the corresponding DPM to calculate counting efficiency.

The absolute amounts of radioisotopes were calculated using Eq. (2):

$$amounts\ (nmoles) = \frac{DPM \times 1000}{specific\ activity\ (\mu Ci\ \mu mole^{-1}) \times 2.22 \times 10^6}. \tag{2}$$

Uptake and incorporation of radioisotopes (hot) into proteins and macromolecules was calculated using Eq. (3):

$$uptake\ or\ incorporation\ (hot,\ nmole\ per\ OD_{600}\ per\ minute) =$$
$$\frac{amounts\ (nmoles)}{labeling\ time\ (min) \times culture\ volume\ (ml) \times OD_{600}}. \tag{3}$$

Incorporation of radioisotopes into nucleic acids was calculated using Eq. (4):

$$incorporation\ (hot,\ nmole\ per\ A_{260}\ per\ minute) =$$
$$\frac{amounts\ (nmoles)}{labeling\ time\ (min) \times nucleic\ acid\ solution\ volume\ (ml) \times A_{260}}. \tag{4}$$

A single yeast cell grown in SD contains ~0.023 pg DNA and ~1.222 pg RNA[64]. One $A_{260}$ unit (1 ml of $A_{260}$ = 1) of DNA = 50 μg and one $A_{260}$ unit of RNA = 40 μg. Under the assumption that one $OD_{600}$ unit contains ~30 million yeast cells[65], one $OD_{600}$ unit of yeast cells contains ~0.93 $A_{260}$ units of nucleic acids. This was used to convert (nmole per $A_{260}$ per minute) to (nmole per $OD_{600}$ per minute). Incorporation of radioisotopes into different RNA classes (mRNA and rRNA) was estimated by assuming mRNA constitutes ~5% and rRNA ~80% of total RNA, respectively[66].

Incorporation of total equivalent amino acids (hot and cold) into macromolecules (nmole per $OD_{600}$ per minute) was approximated using Eq. (5):

$$incorporation\ (total\ equivalents,\ nmole\ per\ OD_{600}\ per\ minute) =$$
$$incorporation\ (hot) + amounts\ of\ cold\ amino\ acid\ (nmole\ per\ OD_{600})$$
$$\times \frac{incorporation\ (hot)}{uptake\ (hot)}. \tag{5}$$

**Analysis of amino acids aminoacylated on tRNAs.** Aminoacylated tRNA was extracted under acidic condition as described[67] and deacylation was performed according to Murthy et al.[68], with some modifications. Briefly, cell pellets were

washed once with ice-cold sodium acetate buffer (0.3 M sodium acetate and 10 mM EDTA, pH 4.5) and resuspended in 600 μl ice-cold sodium acetate buffer and 600 μl acidic phenol (pH 4.3). Cells were lysed with glass beads using a bead beater and the supernatant was extracted again with acidic phenol, followed by a third extraction with chloroform to remove the residual phenol. Nucleic acids were ethanol precipitated, washed with 70% ethanol, resuspended in 10 mM sodium acetate (pH 5.0). Samples were snap frozen in liquid nitrogen and stored at −80 °C. To deacylate aminoacylated tRNAs, we ethanol precipitated total RNA and washed the RNA pellet with 70% ethanol twice to remove the sodium acetate in order to facilitate the subsequent mass spectrometry (MS) analysis. The RNA pellet was resuspended in 30 mM sodium carbonate (pH ~9.0) and incubated at 37 °C for 1 h. Acetonitrile was added to a final concentration of 75% to precipitate nucleic acids and the supernatant was further passed through a 10k filter (Millipore). The flowthrough was sent for MS analysis using the normal-phase method described above.

**Estimation of intracellular amino acid concentration**. Intracellular amino acid concentration was measured for WT cells grown in SD medium and used to extrapolate amino acid concentration for nitrogen-starved WT cells and atg1Δ cells grown with or without nitrogen. Briefly, amino acid concentration was estimated using external calibration curves with standards of known quantities. The intracellular volume of one $OD_{600}$ unit (one ml of $OD_{600} = 1$) is ~1.26 μl (one $OD_{600}$ unit contains ~30 million yeast cells[65] and 42 μm³ is the median intracellular volume[69]) and was used to calculate molar concentrations of amino acids.

**Estimation of protein concentration**. Cell pellets from metabolite extraction were resuspended in 5% SDS (w/v) and 0.1 M NaOH, and protein concentration was estimated using the bicinchoninic acid protein concentration assay kit (Thermo Fisher Scientific) following the manufacturer's instructions. External protein standards were analyzed alongside the samples to estimate protein amounts.

**Western blot**. Cultures were first quenched in 10% ice-cold TCA for 10 minutes on ice and then stored at −80 °C until analysis. Cell pellets were washed once in cold acetone to remove the residual TCA, before bead-beating in urea lysis buffer containing 6 M urea, 1% SDS, 50 mM Tris-HCl pH 7.5, 5 mM EDTA, 1 mM DTT, 1 mM PMSF, 10 μM leupeptin, 5 mM pepstatin A, and 1 × protease inhibitor cocktail (Roche). Lysates were heated for 5 minutes at 75 °C and then centrifuged at maximum speed for 5 minutes. Protein concentration was estimated using the bicinchoninic acid assay (Thermo Fisher Scientific) and equal amounts of proteins were separated by electrophoresis using 4–12% NuPAGE gels. Proteins were then transferred to a nitrocellulose membrane and blotted with the corresponding antibodies. Blocking was performed in 5% dry milk/TBST, while antibody incubation was in 1% dry milk/TBST. Antibodies were used at the following dilutions: α-FLAG 1:3,000 (Sigma F1804 or Cell Signaling #2368S) and α-Rpn10p 1:40,000 (Abcam ab98843).

**Alkaline phosphatase (ALP) assay**. The ALP assay was performed as described with some modifications[70,71]. Briefly, samples were lysed in 250 mM Tris-HCl pH 9.0, 2 mM $MgSO_4$, 1% (v/v) Triton X-100, and 1 × protease inhibitor cocktail (Roche) using glass beads on a bead beater (three cycles of 1-min beating and 1-min cooling on ice). Cell debris and glass beads were removed by centrifugation. Aliquots of 70 μl cell lysates were dispensed into a clear 96-well flat-bottom plate incubated on ice. Reactions were initiated by adding 70 μl substrate solution (250 mM Tris-HCl pH 9.0, 25 mM $MgSO_4$, 1% (v/v) Triton X-100, and 2.7 mM para-nitrophenylphosphate (MP Biomedicals)) and allowed to proceed at room temperature for 5 minutes. Reactions were terminated by adding 140 μl 1 M glycine (pH 11.0). The ALP activity was measured by absorbance at 400 nm ($A_{400}$) and normalized by protein concentration of cell lysates ($A_{595}$ using the Bradford assay (Bio-Rad)). Measurements were performed on a BioTek CYTATION 5 or Synergy HT plate reader.

**Reporting summary**. Further information on research design is available in the Nature Research Reporting Summary linked to this article.

## Data availability
The data that support the findings of this study are within the article and Supplementary Figures. The source data for Figs. 1b, 2b, 3a, 3b, 4a–4c, 5c–5e, 5g, 6b–6d, 7c–7j, 8b–8i, and Supplementary Figs. 1a–1d, 2b, 3a–3c, 4a, 4b, 5a–5c, 6b, 6c, 7b–7d, 8, 9b, 10b–10e, 11, 12a, 12b, 13b, 14b–14f, 15b–15g, 16b, 16c, 17c–17g are provided as a Source Data file. All other data and protocols are available from the corresponding author upon reasonable request. Source data are provided with this paper.

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

## Acknowledgements

We thank Dr. Hamid Baniasadi for assistance with the AB SCIEX QTRAP 6500$^+$. We also thank Dr. Juan Manuel Povedano and members of the Tu lab for critically reading the manuscript and helpful discussions. This work was supported by Welch Foundation I-1797 and NIH R35GM136370 to B.P.T.

## Author contributions

Conceptualization, K.L. and B.P.T.; Methodology, K.L. and B.M.S.; Investigation, K.L. and B.M.S.; Writing—Original draft, K.L. and B.P.T.; Writing—Review and editing, B.M.S.; Funding acquisition, B.P.T.; Supervision, B.P.T.

## Competing interests

The authors declare no competing interests.
