## [Peer Review File · Nature Communications]

REVIEWERS' COMMENTS

Reviewer #1 (Remarks to the Author):

The authors have addressed all of my concerns, and also provide a reasonable response to the concerns raised by the other reviewers. I would note that some of the comments from the other reviewers, in my opinion, were overly critical and at times unfair. In fact, I am disappointed that in some cases their comments raise serious questions about their expertise in metabolism. Of note, I completely disagree with the suggestion that the work in this paper is not novel, as I am not aware of any other publications that demonstrate autophagy only matters for maintaining levels of specific amino acids.

I like the authors edits to the manuscript, including the title, and the paper presents their findings in a fair and balanced way. Without question it should be published without further delay in Nature Communications.

Reviewer #1 (Remarks to the Author):

The authors have addressed all of my concerns, and also provide a reasonable response to the concerns raised by the other reviewers. I would note that some of the comments from the other reviewers, in my opinion, were overly critical and at times unfair. In fact, I am disappointed that in some cases their comments raise serious questions about their expertise in metabolism. Of note, I completely disagree with the suggestion that the work in this paper is not novel, as I am not aware of any other publications that demonstrate autophagy only matters for maintaining levels of specific amino acids.

I like the authors edits to the manuscript, including the title, and the paper presents their findings in a fair and balanced way. Without question it should be published without further delay in Nature Communications.

Response:

We sincerely thank the reviewer for his/her positive evaluation!